# Matrix Estimation for Offline Evaluation in Reinforcement Learning with Low-Rank Structure

**Xumei Xi**     **Christina Lee Yu**
School of Operations Research and Information Engineering
Cornell University
Ithaca, NY 14850
xx269@cornell.edu, cleeyu@cornell.edu

**Yudong Chen**
Department of Computer Sciences
University of Wisconsin-Madison
Madison, WI 53706
yudong.chen@wisc.edu

## Abstract

We consider offline Reinforcement Learning (RL), where the agent does not interact with the environment and must rely on offline data collected using a behavior policy. Previous works provide policy evaluation guarantees when the target policy to be evaluated is covered by the behavior policy, that is, state-action pairs visited by the target policy must also be visited by the behavior policy. We show that when the MDP has a latent low-rank structure, this coverage condition can be relaxed. Building on the connection to weighted matrix completion with non-uniform observations, we propose an offline policy evaluation algorithm that leverages the low-rank structure to estimate the values of uncovered state-action pairs. Our algorithm does not require a known feature representation, and our finite-sample error bound involves a novel discrepancy measure quantifying the discrepancy between the behavior and target policies in the spectral space. We provide concrete examples where our algorithm achieves accurate estimation while existing coverage conditions are not satisfied. Building on the above evaluation algorithm, we further design an offline policy optimization algorithm and provide non-asymptotic performance guarantees.

## 1   Introduction

Reinforcement Learning (RL) has achieved significant empirical success in the online setting, where the agent continuously interacts with the environment to collect data and improve its performance. However, online exploration is costly and risky in many applications, such as healthcare [4] and autonomous driving [16], in which case it is preferable to learn from a pre-collected observational dataset from doctors or human drivers using their own policies. Due to lack of on-policy interaction with the environment, offline RL faces the fundamental challenge of distribution shift [7]. A standard approach for handling distribution shift is importance sampling [12, 11]. More sophisticated approaches have been proposed to alleviate the high variance of importance sampling [2, 21]. Recent works [17, 10, 23] consider estimating the state marginal importance ratio, a more tractable problem.

Existing work on offline RL requires the dataset to have sufficient coverage. A standard measure for coverage is the concentrability coefficient [19]: $C^\pi = \max_{s,a} \frac{d^\pi(s,a)}{\rho(s,a)}$, which is the ratio between the

state-action occupancy measure of a policy $\pi$ of interest and the (empirical) occupancy measure $\rho$ of the behavior policy generating the offline dataset. However, this can be restrictive as the support of $\rho$ must contain that of $d^\pi$ in order for $C^\pi$ to be finite. Earlier work such as the Fitted Q-iteration (FQI) algorithm [9] requires full coverage, i.e. $C^\pi < \infty$ for all policies $\pi$. More recent works [19, 13, 8] requires a more relaxed partial coverage condition $C^{\pi^*} < \infty$ for the optimal policy $\pi^*$. Partial coverage is still a fairly strong requirement: the behavior policy must visit every state the optimal policy would visit, and take every action the optimal policy would take.

In this paper, we seek to relax the coverage condition for offline policy evaluation in settings where the Markov decision process (MDP) has a latent low-rank structure. Similarly to [15, 14], we view the $Q$ function as a matrix and exploit its low-rank structure to infer the entries that were not observed in the offline data. Unlike typical results from the low-rank matrix completion literature, our setting requires completing the matrix under non-uniform sampling, as in [3, 6]; moreover, the error is evaluated under a different distribution or weighted norm, leading to the fundamental challenge of distribution shift. By leveraging techniques from weighted and non-uniform matrix completion, we develop a new offline policy evaluation algorithm, which alternates between Q iteration and matrix estimation. For both the infinite and finite sample settings, we show that the evaluation error can be bounded in terms of a novel discrepancy measure between the behavior and target policies. In contrast to the standard concentrability coefficient, our discrepancy measure may remain finite even when the behavior policy does not cover the support of the target policy. We present a concrete example where the concentrability coefficient is infinite but our method achieves a meaningful error bound. Building on the above evaluation algorithm, we further design an offline policy optimization algorithm with provable performance guarantees.

## 2   Problem Setup

Consider an MDP $\mathcal{M} = (\mathcal{S}, \mathcal{A}, H, P, r, \mu_1)$ with finite state space $\mathcal{S}$, finite action space $\mathcal{A}$, horizon $H$, transition kernel $P = \{P_t\}_{t \in [H]}$, bounded reward function $r = \{r_t : \mathcal{S} \times \mathcal{A} \to [0, 1]\}_{t \in [H]}$, and initial state distribution $\mu_1 \in \Delta(\mathcal{S})$. Let $S = |\mathcal{S}|$ and $A = |\mathcal{A}|$. For each policy $\pi = \{\pi_t : \mathcal{S} \to \Delta(\mathcal{A})\}_{t \in [H]}$, the Q function $Q_t^\pi : \mathcal{S} \times \mathcal{A} \to \mathbb{R}$ is defined as $Q_t^\pi(s, a) = \mathbb{E}_\pi[\sum_{i=t}^H r_i(s_i, a_i)|s_t = s, a_t = a]$, and the total expected reward is $J^\pi = \mathbb{E}_\pi[\sum_{t=1}^H r_t(s_t, a_t)|s_1 \sim \mu_1]$. Let $d_t^\pi : \mathcal{S} \times \mathcal{A} \to [0, 1]$ denote the state-action occupancy measure at time $t \in [H]$ under policy $\pi$.

Given a dataset generated by the behavior policy $\pi^\beta$, our goal is to estimate $J^{\pi^\theta}$ for a target policy $\pi^\theta$. We assume that the MDP has the following low-rank structure, which implies that for any policy $\pi$, its $Q$ function (viewed as an $S$-by-$A$ matrix) is at most rank $d$.

**Assumption.** *For all $t$, $r_t \in [0, 1]^{S \times A}$ has rank at most $d/2$, and $P_t$ admits the decomposition*

$$P_t(s'|s, a) = \sum_{i=1}^{d/2} u_{t,i}(s', s)w_{t,i}(a) \quad or \quad P_t(s'|s, a) = \sum_{i=1}^{d/2} u_{t,i}(s)w_{t,i}(s', a), \quad \forall s', s, a.$$

The above low-rank model is different from the Low-rank MDP model considered in previous works [1, 20]. In Low-rank MDPs, the transition kernel $P$ is assumed to have a factorization of the form $P(s'|s, a) = \sum_{i=1}^d u_i(s')w_i(s, a)$, where the factors $u_i(\cdot)$ and $w_i(\cdot, \cdot)$ are unknown. Closely related is the Linear MDP model [5, 22], where the feature maps $w_i(\cdot, \cdot)$ are known. In these models, the low-rank/linear structures are with respect to the relationship between the originating state-action pair $(s, a)$ and the destination state $s'$; they do *not* imply that $Q$ function is low-rank when viewed as a matrix. In contrast, our model stipulates that the transition kernel can be factorized either between (i) $a$ and $(s, s')$ or (ii) $s$ and $(s', a)$, both of which imply a low dimensional relationship between the current state $s$ and the action $a$ taken at that state, resulting in a low-rank $Q$ function.

For a matrix $M$, let $\|M\|_*$ denote its nuclear norm (sum of singular values), $\|M\|_{\mathrm{op}}$ its operator norm (maximum singular value), $\|M\|_\infty = \max_{i,j} |M_{ij}|$ its entrywise $\ell_\infty$ norm, and $\mathrm{supp}(M) = \{(i, j) : M_{ij} \neq 0\}$ its support. The indicator matrix $\mathbb{1}_M$ is a binary matrix encoding the position of the support of $M$. The entrywise product between two matrices $M$ and $M'$ is denoted by $M \circ M'$.

We propose a novel discrepancy measure defined below, and show that it can replace the role of the concentrability coefficient in our error bound under the low-rank assumption.

**Definition 1** (Operator discrepancy). *The operator discrepancy between two probability distributions $p, q \in \Delta(\mathcal{S} \times \mathcal{A})$ is defined as*

$$\text{Dis}(p\|q) := \min\Big\{ \|g - q\|_{\text{op}} : g \in \Delta(S \times A),\ \text{supp}(g) \subseteq \text{supp}(p) \Big\}. \tag{1}$$

Note that $\text{Dis}(p\|q) \leq \|p - q\|_{\text{op}}$ is always finite, and $\text{Dis}(p\|q) = 0$ if and only if $\text{supp}(q) \subseteq \text{supp}(p)$. To gain intuition for the above definition, assume the minimizer in (1) is $g^*$. By generalized Hölder's inequality, we have

$$\Big| \mathbb{E}_{(s,a) \sim g^*}\big[ M(s,a) \big] - \mathbb{E}_{(s,a) \sim q}\big[ M(s,a) \big] \Big| = \Big| \langle g^*, M \rangle - \langle q, M \rangle \Big| \leq \text{Dis}(p\|q) \cdot \|M\|_*. \tag{2}$$

If the nonzero singular values of $M$ are of the same scale, then the RHS of (2) is of order $\text{Dis}(p\|q) \cdot \text{rank}(M)$. Therefore, $\text{Dis}(p\|q)$ measures the distribution shift between $p$ and $q$ in terms of preserving the expectation of low-rank matrices. Note that $\text{Dis}(p\|q)$ only depends on the support of $p$: if $\text{supp}(p) = \text{supp}(p')$, then $\text{Dis}(p\|q) = \text{Dis}(p'\|q)$ for all $q$. Moreover, thanks to the minimization in the definition (1), $\text{Dis}(p\|q)$ can be significantly smaller than $\|p - q\|_{\text{op}}$. For instance, if $p$ is the uniform distribution on $\mathcal{S} \times \mathcal{A}$, then $g^* = q$ and hence $\text{Dis}(p\|q) = 0$ for all $q$.

The operator discrepancy shares similarity with the parameter $\Lambda$ in [6]. Note that the operator discrepancy is not symmetric. The error bounds for our proposed off-policy evaluation algorithm will be a function of $\text{Dis}(d_t^{\pi^\beta} \| d_t^{\pi^\theta})$, even when one is given infinite samples from the behavior policy. The operator discrepancy only depends on the support of the behavior policy and not the exact distribution, which is expected under the infinite sample setting. As such, the operator discrepancy highlights the inherent error induced by distribution shift. In the finite sample setting, our error will have additional terms which represent the empirical approximation error on the support of the behavior policy.

## 3 Algorithm

Our algorithm alternates between two steps: applying Q-value iteration on the support of $d_t^{\pi^\beta}$, and using low-rank matrix completion to infer the Q values off support. The algorithm takes as input an offline dataset $\mathcal{D} = \{(s_t^k, a_t^k, r_t^k)\}_{t \in [H], k \in [K]}$, which contains $K$ independent trajectories generated from the behavior policy $\pi^\beta$. Over state-action pairs in the support of $\pi^\beta$, we use the data to construct unbiased empirical estimates of the immediate reward, transition kernel and occupancy measure of the behavior policy, denoted by $\widehat{r}_t$, $\widehat{P}_t$ and $\widehat{d}_t^{\pi^\beta}$, respectively. Let $\widehat{B}_t^{\pi^\theta}$ denote the target policy's empirical Bellman operator, which is given by

$$(\widehat{B}_t^{\pi^\theta} f)(s,a) = \widehat{r}_t(s,a) + \sum_{s',a'} \widehat{P}_t(s'|s,a) \pi_t^\theta(a'|s') f(s',a') \tag{3}$$

for all $f : \mathcal{S} \times \mathcal{A} \to \mathbb{R}$. Note that we can evaluate $(\widehat{B}_t^\pi f)(s,a)$ only over $(s,a) \in \text{supp}(\widehat{d}_t^{\pi^\beta})$. With these notations, our algorithm is given below.

---

**Algorithm 1:** Matrix Completion in Low-Rank Offline RL

---

**Data:** dataset $\mathcal{D}$, $\pi^\theta$, initial state distribution $\mu_1$, weight matrices $(\rho_t)_{t \in [H]}$, and $\texttt{ME}(\cdot)$.

**Result:** estimator $\widehat{J}$.

1   $\widehat{Q}_{H+1}^{\pi^\theta}(s,a) \leftarrow 0, \quad \forall (s,a) \in \mathcal{S} \times \mathcal{A}$.

2   **for** *t = H, H-1, …, 1* **do**

3      Q iteration: $Z_t(s,a) \leftarrow (\widehat{B}_t^{\pi^\theta} \widehat{Q}_{t+1}^{\pi^\theta})(s,a)$, for all $(s,a) \in \text{supp}(\rho_t)$.

4      Matrix estimation: $\widehat{Q}_t^{\pi^\theta} \leftarrow \texttt{ME}\left(\mathbb{1}_{\rho_t} \circ Z_t\right)$.

5   **end**

6   Output $\widehat{J} \leftarrow \sum_{s,a} \mu_1(s) \pi_1^\theta(a|s) \widehat{Q}_1^{\pi^\theta}(s,a)$.

---

Line 3 of Algorithm 1 involves a weight matrix $\rho_t$ with $\text{supp}(\rho_t) \subseteq \text{supp}(\widehat{d}_t^{\pi^\beta})$. One may simply take $\rho_t$ to be $\widehat{d}_t^{\pi^\beta}$, or use other weights. Our theorems below quantify the performance of different $\rho_t$.

Line 4 of the algorithm uses a matrix estimation subroutine $\mathtt{ME}(\cdot)$, which is given by the following nuclear norm minimization program with $L_t := H - t + 1$ and a tuning parameter $\kappa \geq 0$:

$$\mathtt{ME}(\mathbb{1}_{\rho_t} \circ Z_t) = \underset{M \in \mathbb{R}^{S \times A}}{\operatorname{argmin}} \|M\|_*$$
$$\text{s.t. } \|\mathbb{1}_{\rho_t} \circ (M - Z_t)\|_\infty \leq \kappa, \quad \|M\|_\infty \leq L_t. \tag{4}$$

## 4  Analysis

We present evaluation error bounds under both the *infinite sample* setting $K \to \infty$ and the *finite sample* setting $K < \infty$. Define the population Bellman operator $B_t^{\pi^\theta}$, which is given by equation (3) with $\widehat{r}_t$ and $\widehat{P}_t$ replaced by $r_t$ and $P_t$. Define the matrix $Y_t \in \mathbb{R}^{S \times A}$ via $Y_t(s, a) = (B_t^{\pi^\theta} \widehat{Q}_{t+1}^{\pi^\theta})(s, a)$, which is the population version of $Z_t$ computed in Algorithm 1.

In the infinite sample setting, we have $\widehat{d}_t^{\pi^\beta}(s, a) \to d_t^{\pi^\beta}(s, a)$, $\widehat{r}_t(s, a) \to r_t(s, a)$ and $\widehat{P}_t(s, a) \to P_t(s, a)$ for all $(s, a) \in \operatorname{supp}(d_t^{\pi^\beta})$. Consequently, both $\widehat{B}_t^{\pi^\theta}$ and $Z_t$ converge to their population versions $B_t^{\pi^\theta}$ and $Y_t$, respectively. We set $\kappa = 0$ in equation (4), which implies that $\mathbb{1}_{\rho_t} \circ M = \mathbb{1}_{\rho_t} \circ Y_t$. We have the following guarantee.

**Theorem 1** (Infinite samples). *In the infinite sample setting, under Algorithm 1 with $\rho_t = d_t^{\pi^\beta}$ and $\kappa = 0$, the output estimator $\widehat{J}$ satisfies*

$$\left|\widehat{J} - J^{\pi^\theta}\right| \leq 2 \sum_{t=1}^{H} \operatorname{Dis}(d_t^{\pi^\beta} \| d_t^{\pi^\theta}) \|Y_t\|_* . \tag{5}$$

Next consider the setting with a finite dataset $\mathcal{D} = \{(s_t^k, a_t^k, r_t^k)\}_{t \in [H], k \in [K]}$. Let $n_t(s, a) := \sum_{k \in [K]} \mathbb{1}_{(s_t^k, a_t^k) = (s, a)}$ be the visitation count of each state-action pair. Accordingly, the empirical occupancy of $\pi^\beta$ is given by $\widehat{d}_t^{\pi^\beta}(s, a) = n_t(s, a)/K$. The following guarantee, which generalizes Theorem 1, holds for any weight matrices $\rho = (\rho_t)_{t \in [H]}$, for which we define $n_{\min}(\rho) := \min\{n_t(s, a) : t \in [H], (s, a) \in \operatorname{supp}(\rho_t)\}$.

**Theorem 2** (Finite samples). *Consider the finite sample setting under Algorithm 1 with $\rho_t$ satisfying* $\operatorname{supp}(\rho_t) \subseteq \operatorname{supp}(\widehat{d}_t^{\pi^\beta})$ *and* $\kappa = \|\mathbb{1}_{\rho_t} \circ (Y_t - Z_t)\|_\infty$ . *There exists an absolute constant $C > 0$ such that with probability at least $1 - \delta$, we have*

$$\left|\widehat{J} - J^{\pi^\theta}\right| \leq 2 \sum_{t=1}^{H} \operatorname{Dis}(\rho_t \| d_t^{\pi^\theta}) \|Y_t\|_* + C \sqrt{\frac{H^3 \log(HSA/\delta)}{n_{\min}(\rho)}}. \tag{6}$$

*Remark* 1. The finite sample error bound in (6) decomposes into infinite sample error and statistical error. The infinite sample error quantifies the intrinsic hardness of distribution shift. The statistical error is due to empirical estimation of the reward and transition kernel. We provide the flexibility of using $\rho_t$ different from $\widehat{d}_t^{\pi^\beta}$, which would allow us to balance the infinite sample error and the statistical error. If $n_{\min}(\widehat{d}^{\pi^\beta})$ is small, the statistical error can be large. In this case, one may choose a $\rho_t$ that puts zero weight on the pairs $(s, a)$ with small count $n_t(s, a)$. The resulting $n_{\min}(\rho)$ can be significantly larger than $n_{\min}(\widehat{d}^{\pi^\beta})$, which reduces the statistical error while potentially increasing the infinite sample error.

The proofs of Theorem 1 and 2 are deferred to Appendix A.1 and A.6, respectively. Our bounds (5) and (6) scale with the operator discrepancy (1), which is always finite, given any behavior policy $\pi^\beta$ and target policy $\pi^\theta$, in contrast to the concentrability coefficient $C^\pi = \max_{s,a} \frac{d^\pi(s,a)}{\rho(s,a)}$. Suppose that there exists some $(s, a)$ such that $\rho_t(s, a) = 0$ and $d_t^{\pi^\theta}(s, a) > 0$. Then, $C^{\pi^\theta} = \infty$ whereas $\operatorname{Dis}(\rho_t \| d_t^{\pi^\theta})$ is finite and meaningful.

We present a concrete example in the infinite samples setting showcasing the effectiveness of our algorithm. Assume $S = A = n$. Consider the simple setting where the transition is uniform over all state-action pairs. For each $s$ and $t$, assume $\pi_t^\theta(\cdot|s)$ is supported on $m$ actions, and the locations of these actions are a realization of uniform random sampling over $[n]$. We assume $\pi_t^\beta$ is generated from the same model independently. Note that the support of $d_t^{\pi^\beta}$ and $d_t^{\pi^\theta}$ will be mostly disjoint, making the concentrability coefficient infinite with high probability. Using Theorem 1, we derive the following error bound, the proof of which is deferred to Appendix A.2.

**Corollary 1.** *Under the aforementioned setting, there exists an absolute constant $C > 0$ such that when $n \geq C$, with probability at least $1 - \frac{1}{n}$, we have $\left| \widehat{J} - J^{\pi^\theta} \right| \leq C\sqrt{\frac{dH^3 \log(n)}{m}}$.*

If $m$ satisfies $m \gtrsim \frac{dH \log n}{\epsilon^2}$, then we have $|\widehat{J} - J^{\pi^\theta}| \leq \epsilon H$. Suppose $m = n/2$. In this setting, the behavior and target policies both randomize over half of the actions, but their actions may have little overlap. Our bound gives $|\widehat{J} - J^{\pi^\theta}| \lesssim \sqrt{\frac{dH^3 \log n}{n}}$, which can be vanishingly small when $n$ is large.

# 5 Policy Optimization and Guarantee

In this section, we build on our policy evaluation methode to design an offline policy optimization algorithm. Given a dataset $\mathcal{D}$ generated by a behavior policy $\pi^\beta$, we can use Algorithm 1 to obtain an value estimate $\widehat{J}^\pi$ for each policy $\pi$. We optimize over policies for which the above estimate is reliable. Specifically, we consider the following set of candidate policies

$$\Pi_B := \left\{ \pi : \mathrm{Dis}(d_t^\pi \| \rho_t) \leq B, \forall t \in [H] \right\},$$

where the parameter $B \geq 0$ controls how close the occupancy distribution of the candidate policy is to the data distribution, in terms of operator discrepancy defined in (1). It is easy to see that when policy $\pi$ satisfies $\mathrm{supp}(d_t^\pi) \subseteq \mathrm{supp}(\rho_t)$, we have $\mathrm{Dis}(d_t^\pi \| \rho_t) = 0$ and as a result, $\pi \in \Pi_B$ for any $B \geq 0$. Consequently, if we take $\rho_t = \widehat{d}_t^{\pi^\beta}$, then $\Pi_B$ includes the policies that are covered by the offline dataset. In other words, all policies with finite concentrability coefficients are in $\Pi_B$. Importantly, when $B > 0$, the set $\Pi_B$ contains other policies with infinite concentrability coefficients, as demonstrated in the example at the end of last section. With a bigger $B$, the set $\Pi_B$ includes more policies, though we have weaker evaluation guarantees for these policies.

Among all candidate policies in $\Pi_B$, we maximize the estimated values obtained by Algorithm 1 to get

$$\widehat{\pi} = \underset{\pi \in \Pi_B}{\mathrm{argmax}} \; \widehat{J}^\pi. \tag{7}$$

We present the following guarantee for $\widehat{\pi}$, the proof of which can be found in Appendix A.7.

**Theorem 3.** *Suppose $\pi^\beta \in \Pi_B$. We obtain $\widehat{\pi}$ by solving (7). There exists an absolute constant $C > 0$ such that with probability at least $1 - \delta$, we have*

$$J^{\widehat{\pi}} \geq J^\pi - 4BH^{3/2}\sqrt{SAd} - C\sqrt{\frac{H^3 \log(HSA/\delta)}{n_{\min}(\rho)}}, \quad \forall \pi \in \Pi_B. \tag{8}$$

The above bound shows that we are able to find a policy $\widehat{\pi}$ with a nearly optimal value, compared to other policies in $\Pi_B$. How close $\widehat{\pi}$ is to the optimal policy in $\Pi_B$ depends on how accurately we can evaluate all policies in $\Pi_B$. According to Theorem 2, the estimations are accurate if $B$ is small (policies are close to behavior) and $n_{\min}(\rho)$ is large (dataset is large), which is reflected in the bound (8). Similarly as before, the two error terms in (8) quantifies the fundamental difficulty of distribution shift and finite sample noise, respectively. If we let $\rho_t = \widehat{d}_t^{\pi^\beta}$ and $B = 0$, we are essentially optimizing over all policies with finite concentrability coefficients. In this case, our bound reduces to

$$J^{\widehat{\pi}} \geq J^\pi - C\sqrt{\frac{H^3 \log(HSA/\delta)}{n_{\min}(\rho)}}, \quad \forall \pi \in \Pi_0.$$

Compared to the PAC bound of CPPO proposed in [19] (Section 5.1), our bound has a better polynomial dependence on the time horizon $H$.

# 6 Conclusion

We propose a novel algorithm for efficient offline evaluation when low-rank structure is present in the MDP. Our algorithm is a combination of Q iteration and low-rank matrix estimation, which is easy to implement. We show that the proposed operator discrepancy measure better captures the difficulty of policy evaluation in the offline setting, compared to the traditional concentrability coefficient. We

also combine the evaluation algorithm with policy optimization and provide performance guarantee. We believe that this work is a first step in exploiting the benefit of low-rank structure in the Q function in offline RL. In the future, we hope to develop a more efficient policy optimization algorithm, with a better estimation accuracy.

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

# A   Proofs

Let $\mu_t^\pi : \mathcal{S} \to [0,1]$ denote the state occupancy measure at time $t \in [H]$ under policy $\pi$.

## A.1   Proof of Theorem 1

We present two lemmas before analyzing the evaluation error. The first one analyzes the error incurred at the matrix estimation step. The proof is deferred to Appendix A.4.

**Lemma 1.** *For arbitrary real matrices $A, B, P, W \in \mathbb{R}^{m \times n}$, we have*

$$\left| \sum_{i,j} W_{ij}(A_{ij} - B_{ij}) \right| \leq \left| \sum_{i,j} P_{ij}(A_{ij} - B_{ij}) \right| + (\|A\|_* + \|B\|_*) \|P - W\|_{\mathrm{op}}.$$

*Remark.* Under the matrix estimation framework, we can interpret matrix $P$ as the sampling pattern and $W$ as the weights for evaluation.

Next, we introduce a lemma decomposing the evaluation error as a summation of the matrix estimation accuracy from future timesteps. The proof can be found in Appendix A.5.

**Lemma 2.** *For the Q function and its estimator $Q_t^{\pi^\theta}, \widehat{Q}_t^{\pi^\theta} \in \mathbb{R}^{S \times A}$, we have*

$$\left\langle d_t^{\pi^\theta}, \widehat{Q}_t^{\pi^\theta} - Q_t^{\pi^\theta} \right\rangle = \left\langle d_t^{\pi^\theta}, \widehat{Q}_t^{\pi^\theta} - Y_t \right\rangle + \left\langle d_{t+1}^{\pi^\theta}, \widehat{Q}_{t+1}^{\pi^\theta} - Q_{t+1}^{\pi^\theta} \right\rangle, \; t \in [H],$$

*and consequently*

$$\left\langle d_1^{\pi^\theta}, \widehat{Q}_1^{\pi^\theta} - Q_1^{\pi^\theta} \right\rangle = \sum_{t=1}^{H} \left\langle d_t^{\pi^\theta}, \widehat{Q}_t^{\pi^\theta} - Y_t \right\rangle.$$

Based on Lemma 1 and 2, we derive the following error bound. For each $t \in [H]$ and an arbitrary $g \in \Delta(S \times A)$ with $\mathrm{supp}(g) \subseteq \mathrm{supp}(d_t^{\pi^\beta})$, we have

$$\left| \sum_{s,a} d_t^{\pi^\theta}(s,a) \left( \widehat{Q}_t^{\pi^\theta}(s,a) - Y_t(s,a) \right) \right|$$

$$\leq \left| \sum_{s,a} g(s,a) \left( \widehat{Q}_t^{\pi^\theta}(s,a) - Y_t(s,a) \right) \right| + 2 \|Y_t\|_* \left\| d_t^{\pi^\theta} - g \right\|_{\text{op}}$$

$$= 2 \|Y_t\|_* \left\| d_t^{\pi^\theta} - g \right\|_{\text{op}},$$

where the first step follows from Lemma 1, and the second equality follows from the constraints in (4) with $\kappa = 0$. Then, combining with the decomposition in Lemma 2, we obtain the desired bound by minimizing over all such $g$.

## A.2  Proof of Corollary 1

For simplicity, define $b := \frac{m}{n}$. Since the transition is uniform, the state occupancy $\mu_t^\pi(\cdot)$ is uniform under any policy $\pi$, i.e. $\mu_t^\pi(s) = \frac{1}{n}$. By the way the policies are generated, $d_t^{\pi^\theta}(\cdot, \cdot) = \mu_t^{\pi^\theta}(\cdot)\pi_t^\theta(\cdot|\cdot) \in \mathbb{R}^{n \times n}$ is supported on $mn$ entries whose locations are realization of random sampling, and on these entries $d_t^{\pi^\theta}(s,a) = \frac{1}{mn}$. Specifically, all $d_t^{\pi^\theta}(s,a)$ are i.i.d. Bernoulli random variables that take the value 1 with probability $b$. The behavior policy $\pi^\beta$ is generated independently via the same process. Let $M := d_t^{\pi^\theta} - d_t^{\pi^\beta}$ and we have

$$M_{ij} = \begin{cases} \frac{1}{mn} & \text{with probability } b(1-b) \\ -\frac{1}{mn} & \text{with probability } b(1-b) \\ 0 & \text{with probability } 1 - 2b(1-b) \end{cases} \tag{9}$$

independently across all entries $(i,j)$. By matrix Bernstein inequality, we obtain the following result, the proof of which is deferred to Appendix A.3.

**Lemma 3.** *There exists an absolute constant $C > 0$ such that when $n \geq C$, with probability at least $1 - \frac{1}{n}$, we have*

$$\|M\|_{\text{op}} \leq C \frac{1}{n} \sqrt{\frac{\log(n)}{m}}. \tag{10}$$

Since $Y_t$ is at most rank-$d$ and has bounded entries, we have $\|Y_t\|_* \leq \sqrt{d} \|Y_t\|_F \leq n\sqrt{dH}$. Combining Theorem 1 and (10), we get

$$\left| \widehat{J} - J^{\pi^\theta} \right| \leq 2 \sum_{t=1}^H \|Y_t\|_* \left\| d_t^{\pi^\beta} - d_t^{\pi^\theta} \right\|_{\text{op}}$$

$$\lesssim H \cdot n\sqrt{dH} \cdot \frac{1}{n} \sqrt{\frac{\log(n)}{m}}$$

$$= \sqrt{\frac{dH^3 \log(n)}{m}},$$

where the first upper bound is obtained by plugging $d_t^{\pi^\beta}$ into the objective of (1).

## A.3  Proof of Lemma 3

We apply matrix Berstein's inequality (Theorem 6.1.1 in [18]). Let $S_k = M_{ij} e_i e_j^\top$, for all $k \in [n^2]$. Since $|M_{ij}| \leq \frac{1}{mn}$, we derive that $\|S_k\|_{\text{op}} \leq \frac{1}{mn}$. We calculate that

$$\sum_k \mathbb{E}[S_k S_k^\top] = \sum_{i,j} \mathbb{E}[M_{ij}^2] e_i e_i^\top$$

$$= 2b(1-b) \frac{1}{m^2 n} I_n.$$

As a result, we have

$$\left\| \sum_k \mathbb{E}[S_k S_k^\top] \right\|_{\text{op}} = 2b(1-b) \frac{1}{m^2 n} = \frac{2(n-m)}{n^3 m} \leq \frac{2}{n^2 m}.$$

By symmetry, we also have $\left\|\sum_k \mathbb{E}[S_k^\top S_k]\right\|_{\mathrm{op}} \leq \frac{2}{n^2 m}$. Hence, we get

$$\mathbb{P}\left(\|M\|_{\mathrm{op}} \geq t\right) \leq 2n \exp\left(\frac{-t^2/2}{\frac{2}{mn^2} + \frac{t}{3mn}}\right).$$

Letting the RHS be upper bounded by $\frac{1}{n}$ yields the desired result.

### A.4  Proof of Lemma 1

The proof uses the following result, which holds for any pairs of dual norms. In this paper, we only consider using $\|\cdot\|_*$ and $\|\cdot\|_{\mathrm{op}}$.

**Lemma 4.** *For a real matrix $M \in \mathbb{R}^{m \times n}$ and two weight matrices $P, W \in \mathbb{R}^{m \times n}$, we have that*

$$\left| \sum_{i,j} P_{ij} M_{ij} - \sum_{i,j} W_{ij} M_{ij} \right| \leq \|M\|_* \|P - W\|_{\mathrm{op}}.$$

*Proof.* We can rewrite $\sum_{i,j} P_{ij} M_{ij} - \sum_{i,j} W_{ij} M_{ij}$ as

$$\langle M, P - W \rangle,$$

where $\langle \cdot, \cdot \rangle$ denotes the trace inner product between matrices. Applying Hölder's inequality, we obtain

$$|\langle M, P - W \rangle| \leq \|M\|_* \|P - W\|_{\mathrm{op}}.$$

$\square$

Substituing $M_{ij} = A_{ij} - B_{ij}$ in Lemma 4, we immediately obtain the desired results in Lemma 1.

### A.5  Proof of Lemma 2

Recall that

$$Q_t^{\pi^\theta}(s,a) = \left(B_t^{\pi^\theta} Q_{t+1}^{\pi^\theta}\right)(s,a), \tag{11}$$

$$Y_t(s,a) = \left(B_t^{\pi^\theta} \widehat{Q}_{t+1}^{\pi^\theta}\right)(s,a). \tag{12}$$

For each $(s,a) \in \mathcal{S} \times \mathcal{A}$, we have

$$\begin{aligned}
&\widehat{Q}_t^{\pi^\theta}(s,a) - Q_t^{\pi^\theta}(s,a) \\
&= \left(\widehat{Q}_t^{\pi^\theta}(s,a) - Y_t(s,a)\right) + \left(Y_t(s,a) - Q_t^{\pi^\theta}(s,a)\right) \\
&= \left(\widehat{Q}_t^{\pi^\theta}(s,a) - Y_t(s,a)\right) + \sum_{s',a'} P_t(s'|s,a)\pi_{t+1}^\theta(a'|s')\left(\widehat{Q}_{t+1}^{\pi^\theta}(s',a') - Q_{t+1}^{\pi^\theta}(s',a')\right),
\end{aligned}$$

where the last step follows from equations (12) and (11). Multiplying both sides by $d_t^{\pi^\theta}(s,a)$ and summing over $(s,a)$, we obtain

$$\begin{aligned}
&\left\langle d_t^{\pi^\theta}, \widehat{Q}_t^{\pi^\theta} - Q_t^{\pi^\theta} \right\rangle \\
&= \left\langle d_t^{\pi^\theta}, \widehat{Q}_t^{\pi^\theta} - Y_t \right\rangle + \sum_{s',a'} \underbrace{\sum_{s,a} d_t^{\pi^\theta}(s,a) P_t(s'|s,a) \pi_{t+1}^\theta(a'|s')}_{= d_{t+1}^{\pi^\theta}(s',a')} \left(\widehat{Q}_{t+1}^{\pi^\theta}(s',a') - Q_{t+1}^{\pi^\theta}(s',a')\right) \\
&= \left\langle d_t^{\pi^\theta}, \widehat{Q}_t^{\pi^\theta} - Y_t \right\rangle + \left\langle d_{t+1}^{\pi^\theta}, \widehat{Q}_{t+1}^{\pi^\theta} - Q_{t+1}^{\pi^\theta} \right\rangle,
\end{aligned}$$

thereby proving the first equation in the lemma. Continuing the above recursion yields the second equation.

## A.6  Proof of Theorem 2

Fix $t \in [H]$. For $g_t \in \Delta(S \times A)$ satisfying $\mathrm{supp}(g_t) \subseteq \mathrm{supp}(\rho_t)$, we have

$$\left| \sum_{s,a} d_t^{\pi_\theta}(s,a) \left( \widehat{Q}_t^{\pi_\theta}(s,a) - Y_t(s,a) \right) \right|$$

$$\leq \left| \sum_{s,a} g_t(s,a) \left( \widehat{Q}_t^{\pi_\theta}(s,a) - Y_t(s,a) \right) \right| + 2 \left\| Y_t \right\|_* \cdot \left\| d_t^{\pi_\theta} - g_t \right\|_{\mathrm{op}},$$

by Lemma 1 and the fact that $\left\| \widehat{Q}_t^{\pi^\theta} \right\|_* \leq \left\| Y_t \right\|_*$ by construction. Applying triangle inequality, we get

$$\left| \sum_{s,a} g_t(s,a) \left( \widehat{Q}_t^{\pi_\theta}(s,a) - Y_t(s,a) \right) \right|$$

$$\leq \left| \sum_{s,a} g_t(s,a) \left( \widehat{Q}_t^{\pi_\theta}(s,a) - Z_t(s,a) \right) \right| + \left| \sum_{s,a} g_t(s,a) \left( Z_t(s,a) - Y_t(s,a) \right) \right|$$

$$\leq 2 \left| \sum_{s,a} g_t(s,a) \left( Z_t(s,a) - Y_t(s,a) \right) \right|,$$

where the last step follows from the constraint in (4). Recall that

$$Z_t(s,a) = \widehat{r}_t(s,a) + \sum_{s',a'} \widehat{P}_t(s'|s,a) \pi_{t+1}^\theta(a'|s') \widehat{Q}_{t+1}^{\pi_\theta}(s',a'),$$

$$Y_t(s,a) = r_t(s,a) + \sum_{s',a'} P_t(s'|s,a) \pi_{t+1}^\theta(a'|s') \widehat{Q}_{t+1}^{\pi_\theta}(s',a').$$

Using the above expressions, we obtain

$$Z_t(s,a) - Y_t(s,a) = \widehat{r}_t(s,a) - r_t(s,a) + \underbrace{\sum_{s',a'} (\widehat{P}_t(s'|s,a) - P_t(s'|s,a)) \pi_{t+1}^\theta(a'|s') \widehat{Q}_{t+1}^{\pi^\theta}(s',a')}_{\widehat{F}_t(s,a)}.$$

Note that $\widehat{r}_t$ and $\widehat{F}_t$ are both the average of $n_t(s,a)$ bounded random variables. Additionally, $|\widehat{r}_t|$ is bounded by 1 and $\left| \widehat{F}_t \right|$ is bounded by $H$. A standard application of Hoeffding's inequality followed by union bound yields

$$\mathbb{P}\left[ \max_{t,(s,a) \in \mathrm{supp}(\rho_t)} |\widehat{r}_t(s,a) - r_t(s,a)| \leq \varepsilon \right] \geq 1 - \sum_{t,(s,a) \in \mathrm{supp}(\rho_t)} e^{-\varepsilon^2 n_t(s,a)/2}$$

$$\geq 1 - HSA e^{-\varepsilon^2 n_{\min}(\rho)/2}.$$

Setting the probability to be lower bounded by $1 - \delta$ gives

$$\varepsilon \lesssim \sqrt{\frac{\log(HSA/\delta)}{n_{\min}(\rho)}}.$$

Similar techniques yield

$$\mathbb{P}\left[ \max_{t,(s,a) \in \mathrm{supp}(\rho_t)} \left| \widehat{F}_t(s,a) \right| \lesssim \sqrt{\frac{H \log(HSA/\delta)}{n_{\min}(\rho)}} \right] \geq 1 - \delta.$$

Applying the high probability bound and Lemma 2, we obtain

$$\sum_{t=1}^H \left| \sum_{s,a} d_t^{\pi_\theta}(s,a) \left( \widehat{Q}_t^{\pi_\theta}(s,a) - Y_t(s,a) \right) \right|$$

$$\lesssim \sum_{t=1}^H \left\| g_t - d_t^{\pi^\theta} \right\|_{\mathrm{op}} \left\| Y_t \right\|_* + \sqrt{\frac{H^3 \log(HSA/\delta)}{n_{\min}(\rho)}}.$$

Minimizing the RHS over all $g_t$ yields the desired result.

*Remark* 2. Note that the proof does not rely on the specific distribution of $\rho_t$. Instead, we only use the support of $\rho_t$ and upper bound the statistical error by the maximum deviation on the support. It will be an interesting future direction to incorporate the exact distribution of $\rho_t$ into the error bound.

### A.7 Proof of Theorem 3

Invoking Theorem 2, we have

$$
\begin{aligned}
J^{\widehat{\pi}} &\geq \widehat{J}^{\widehat{\pi}} - 2\sum_{t=1}^{H} \mathrm{Dis}(\rho_t \| d_t^{\widehat{\pi}}) \left\| Y_t^{\widehat{\pi}} \right\|_* - C\sqrt{\frac{H^3 \log(HSA/\delta)}{n_{\min}(\rho)}} \\
&\geq \widehat{J}^{\pi} - 2\sum_{t=1}^{H} \mathrm{Dis}(\rho_t \| d_t^{\widehat{\pi}}) \left\| Y_t^{\widehat{\pi}} \right\|_* - C\sqrt{\frac{H^3 \log(HSA/\delta)}{n_{\min}(\rho)}} \\
&\geq J^{\pi} - 2\sum_{t=1}^{H} \mathrm{Dis}(\rho_t \| d_t^{\widehat{\pi}}) \left\| Y_t^{\widehat{\pi}} \right\|_* - 2\sum_{t=1}^{H} \mathrm{Dis}(\rho_t \| d_t^{\pi}) \left\| Y_t^{\pi} \right\|_* - 2C\sqrt{\frac{H^3 \log(HSA/\delta)}{n_{\min}(\rho)}},
\end{aligned}
$$

with probability at least $1 - \delta$. Since we assume $\mathrm{Dis}(\rho_t \| d_t^{\pi}) \leq B$ for all $t \in [H]$ and $\pi \in \Pi_B$, the discrepancies can be upper bounded by $B$. Combining with the fact that $\| Y_t^{\pi} \|_* \leq \sqrt{d} \| Y_t^{\pi} \|_F \leq \sqrt{SAHd}$, we get the desired result.