# OpenReview forum: "Matrix Estimation for Offline Evaluation in Reinforcement Learning with Low-Rank Structure"
_NeurIPS.cc/2022/Workshop/Offline_RL — Offline RL Workshop NeurIPS 2022_

### Official Review · Reviewer_xxoZ · 2022-10-20
**Paper studies an interesting setting, but have concerns about the novelty of its contribution**

**Rating:** 5
**Confidence:** 3

**Review:**

The authors study what happens when the MDP's transition model admits a low-rank factorization. There, the authors show that typical assumptions made in offline evaluation on the support of the behavior and evaluated policy can be relaxed. The authors also propose an offline RL algorithm that uses matrix completion to infer the Q-values of unseen state-action pairs.

Overall, I think the paper is sound and well-written. The setting of low-rank MDPs is also interesting to study from a theoretical standpoint. However, though the authors claim their paper to be a "first step," I am aware of a couple other papers that also study offline RL in low-rank MDPs [1, 2]. Specifically [2] proposes an algorithm that is similar to the one the authors consider. The primary difference between the two is that [2] estimates pessimistic Q-values, which is more common in offline RL algorithms. By not estimating pessimistic values, the analysis is also greatly simplified (as unlike linear MDPs, pessimism in low-rank MDPs is quite difficult to achieve because the low-rank features are not known). I think the paper could benefit from a detailed discussion of how this work differs from these papers, as it is currently not clear to me.

[1] https://arxiv.org/pdf/2006.10814.pdf
[2] https://arxiv.org/pdf/2110.04652.pdf (Section 5 in particular)